# Nanopore-Sequencing Characterization of the Gut Microbiota of *Melolontha melolontha* Larvae: Contribution to Protection against Entomopathogenic Nematodes?

**DOI:** 10.3390/pathogens10040396

**Published:** 2021-03-25

**Authors:** Ewa Sajnaga, Marcin Skowronek, Agnieszka Kalwasińska, Waldemar Kazimierczak, Karolina Ferenc, Magdalena Lis, Adrian Wiater

**Affiliations:** 1Centre for Interdisciplinary Research, Laboratory of Biocontrol, Application and Production of EPN, John Paul II Catholic University of Lublin, Konstantynów 1J, 20-708 Lublin, Poland; marskow@kul.pl (M.S.); waldemar.kazimierczak@kul.pl (W.K.); mdybala@kul.pl (M.L.); 2Department of Environmental Microbiology and Biotechnology, Nicolaus Copernicus University in Toruń, Lwowska 1, 87-100 Toruń, Poland; kala@umk.pl; 3Centre for Biomedicine Research, Warsaw University of Life Sciences, Nowoursynowska 100, 02-797 Warsaw, Poland; karolina_ferenc@sggw.pl; 4Department of Industrial and Environmental Microbiology, Institute of Biological Sciences, Maria Curie-Skłodowska University, Akademicka 19, 20-033 Lublin, Poland; adrianw2@poczta.umcs.lublin.pl

**Keywords:** *Melolontha melolontha*, entomopathogenic nematodes, gut microbiota, host protection, metataxonomics, *Xenorhabdus*, Photorhabdus, pest biocontrol

## Abstract

This study focused on the potential relationships between midgut microbiota of the common cockchafer *Melolontha melolontha* larvae and their resistance to entomopathogenic nematodes (EPN) infection. We investigated the bacterial community associated with control and unsusceptible EPN-exposed insects through nanopore sequencing of the 16S rRNA gene. Firmicutes, Proteobacteria, Actinobacteria, and Bacteroidetes were the most abundant bacterial phyla within the complex and variable midgut microbiota of the wild *M. melolontha* larvae. The core microbiota was found to include 82 genera, which accounted for 3.4% of the total number of identified genera. The EPN-resistant larvae differed significantly from the control ones in the abundance of many genera belonging to the Actinomycetales, Rhizobiales, and Clostridiales orders. Additionally, the analysis of the microbiome networks revealed different sets of keystone midgut bacterial genera between these two groups of insects, indicating differences in the mutual interactions between bacteria. Finally, we detected *Xenorhabdus* and *Photorhabdus* as gut residents and various bacterial species exhibiting antagonistic activity against these entomopathogens. This study paves the way to further research aimed at unravelling the role of the host gut microbiota on the output of EPN infection, which may contribute to enhancement of the efficiency of nematodes used in eco-friendly pest management.

## 1. Introduction

The insect gut is an organ with a highly diverse structure and a rich symbiotic community, which contains mostly bacteria but also archaea, fungi, viruses, and protozoa [1,2]. The composition of insect gut microbiota has been extensively explored with the use of high-throughput DNA sequencing techniques over the last decade, circumventing the limitations of culture-based methods [3]. It seems that the diet, habitat, host taxonomy, and developmental stage determine gut assemblies most significantly [4,5]. Intestine residents can confer a wide array of advantages for the host through involvement in e.g., nutrition, development, communication, or detoxification. They can also protect their hosts against pathogens and parasites, which can be achieved indirectly by enhancing the insect innate immune system or intestinal epithelium cell regeneration and directly by competition for nutrients and space or production of antagonist compounds [1,6,7]. Defensive bacteria have been found in the microbiota of many insect species [8,9,10,11]. Based on their diverse abilities, the potential application of insect gut bacteria in medicine and industry is being recognized [12,13,14].

The common cockchafer *Melolontha melolontha* (Coleoptera: Scarabaeidae) is one of the widely distributed scarab species in Euro-Asia. Its larvae go through three instars, spending up to four years underground, after which pupation and metamorphosis take place, which is linked to changing the feeding habits from rhizophagous to grazing [15]. Larvae of *M. melolontha* and other related scarabs can be extremely destructive in a wide range of crops during occasional population outbreaks [16]. They are regarded as difficult to control, as they take cryptic positions in the soil and exploit a variety of niches. Nevertheless, effective biocontrol agents, such as entomopathogenic nematodes (EPN) against scarab pests have been developed [17,18,19]. These EPN from the genera *Steinernema* and *Heterorhabditis* require intestinal mutualistic *Xenorhabdus* and *Photorhabdus* spp. bacteria (referred to as EPN bacteria) to go through their life cycle [20,21]. EPN infective juveniles (IJ) gain access to the insect digestive tract through natural openings, perforate the midgut wall, and release their mutualists to hemolymph. *Xenorhabdus* and *Photorhabdus* bacteria are able to kill the insect host rapidly by producing a wide range of natural products, allowing propagation of both symbiotic partners. When nutrients are depleted, IJ specifically reassociate with their mutualistic bacteria and then disperse in soil until they find a new insect host to invade [22].

The insect-EPN interactions have been studied intensively for the last two decades in investigations focused mainly on the infection mechanisms, insect immunity-based resistance to EPN, and the secretory capacity of EPN bacteria. However, little is known of the interaction between the gut microbiota and EPN or their mutualistc bacteria during infection. Such interactions are expected to be competitive, as *Xenorhabdus* and *Photorhabdus* bacteria can produce antimicrobial factors to inhibit the growth of gut bacteria [23], while some gut microbiota displays opposite action in vitro [24]. On the other hand, EPN infection promotes the translocation of some gut bacteria into the hemolymph together with nematodes, where they are able to reproduce, competing with EPN bacteria via secretion of antimicrobials [25]. Profiling the bacterial community present in larval cadavers revealed that different bacterial strains from the host microbiota can contribute directly or indirectly to insect death provoked by EPN and their microsymbionts and then share the insect cadaver or even dominate the bacterial community [26,27].

In our previous research, we identified bacteria from the *M. melolontha* larva midgut exhibiting antagonistic activity against bacterial symbionts of EPN, which gave rise to the hypothesis that they can actively participate in defense against EPN infection in scarab species [24]. In this study, we focus on: (i) profiling the midgut bacterial community of wild *M. melolontha* larvae using high-throughput sequencing; (ii) comparing the bacterial microbiota of unsusceptible EPN-exposed individuals to control ones to address the question of whether the differences in the resistance to EPN of some larvae could be attributed, at least partially, to changes in the structure of their gut bacterial community; and (iii) determining the relative abundance of bacterial antagonists of *Xenorhabdus* and *Photorhabdus* spp. isolated earlier from, attempting to establish whether the increased resistance of *M. melolontha* larvae to EPN infection is correlated with the presence of these bacteria in the midgut.

Metataxonomics is a powerful tool for comprehensive characterization of changes in the gut microbial diversity during pathogen infection, also providing information on the association of gut bacteria with host protection [3]. We examined the diversity of the midgut bacterial community using nanopore sequencing of the 16S rRNA gene, which offers a great advantage over other next-generation sequencing platforms by providing relatively long reads spanning almost the whole 16S rRNA gene [28].

## 2. Results

### 2.1. Nanopore Sequencing Results

We characterized the midgut bacterial communities of 27 *M. melolontha* L2 and L3 larva samples collected from soils of eastern Poland, of which 13 larvae were selected in the laboratory as EPN- resistant (Table 1, Appendix A). The midgut microbiota of the larvae were investigated using fourth-generation sequencing of the 16S rRNA gene with the nanopore technology. The bioinformatic data processing resulted in 6,073,855 high quality reads. The number of bacterial reads obtained across all the samples ranged from 77,758 to 675,822 (median 185,006). The rarefraction curves indicated a high coverage of bacterial diversity in the insect midguts (Appendix A). The taxonomic classification rate varied from 88.8 to 97.9% (median 93.9%) and from 69.2 to 94.1% (median 88.3%) at the genus and species level, respectively (Appendix A).

### 2.2. Bacterial Species Richness and α-Diversity

A high level of bacterial α-diversity was observed in each individual studied, as indicated by Sobs (range 595–1217; median 865), Shannon-Wiener (range H’ 2.4–5.1; median 4.2), and Simpson indices (range 1-D 0.67–0.98, median 0.96). The studied communities exhibited quite equal representation (E 0.13–0.15, median 0.14) (Appendix A). On average, the midgut of an individual larva harbored 30 different bacterial phyla (range 24–35), 63 classes (46–760), 320 families (265–380), 1125 genera (866–1834), and 3411 species (1045–6049). The whole dataset contained 2402 genera and 11,532 species; 82 genera were shared across the studied midgut samples, constituting 3.4% of the total number of genera (2403). These genera were considered as the core midgut bacterial community.

### 2.3. Diversity of Bacterial Microbiota of Control Group of M. melolontha Larvae

The most abundant bacterial phyla indicated by relative abundance (RA) values averaged in the midgut of the control group of larvae were as follows: Firmicutes (72.9%), Proteobacteria (12.2%), Bacteroidetes (6.8%), and Actinobacteria (5.9%) (Table 2). The other detected phyla showed RA < 1%. At the class level, the studied bacterial communities were represented mainly by Clostridia (49.4%), Bacilli (12.7%), Erysipelotrichia (7.8%), γ-Proteobacteria (6.9%), Bacteroidia (6.6%), Actinobacteria (5.9%), α-Proteobacteria (3.1%), and β-Proteobacteria (1.1%) (Table 2). The other detected classes showed RA < 1%. The analysis of the control larvae communities at the family level revealed 14 families with RA > 1%, with the most abundant *Lachnospiraceae* (21.4%), *Ruminococcaceae* (18.0%), *Erysipelotrichaceae* (7.8%), *Bacteroidacae* (5.0%), and *Bacillaceae* (4.1%) (Table 3). The *M. melolontha* midgut microbiota of the control group were represented mainly by the following bacterial genera: *Turicibacter* (7.4%), *Lachnoclostridium* (5.9%), *Bacteroidetes* (5.2%), *Anaerotruncus* (5.2%), *Bacillus* (3.2%), *Paenibacillus* (3.0%), and *Ruminococcus* (2.3%) (Appendix A). The other detected genera displayed RA < 2%

The comparison of the structure of the midgut bacterial community across the *M. melolontha* larvae studied demonstrated that the abundance of the detected taxa varied significantly. As many as 94.5% of all the detected bacterial genera were not shared by all larvae and 38.0%, on average, were specific for single larvae. The intrasample variability was also evident in the PCoA analysis (Figure 1).

To assess the effect of the larval developmental stage on the midgut bacterial diversity, samples from *M. melolontha* L2 and L3 larvae were compared. The gut microbiota of the L2 and L3 instar of the control larvae did not differ from each other in terms of richness and evenness (Appendix A). There was also no obvious clustering based on the genus composition and developmental stage of the larvae in the PCoA ordination (Figure 1), and the ANOSIM analysis gave no statistical support for the dissimilarity (R = −0.0562, *p* = 0.699), indicating that the L2 and L3 larvae did not have distinct bacterial microbiota.

### 2.4. Selection of EPN-Resistant Insects

Since *M. melolontha* larvae are relatively resistant to EPN infection, we used a large dose of EPN (1000 IJ/1 insect) for *M. melolontha* larvae infestation. This allowed separation of individuals with the highest EPN resistance from susceptible ones. We found that the insect survival rate after infestation of a group of larvae with *H. megidis*, *S. arenarium, S. bicornutum, S. carposapsae,* and *S. kraussei* at 20 °C for 12 days was 57%, 45%, 47%, 75%, and 65%, respectively.

### 2.5. Diversity of the Bacterial Microbiota of the EPN-Resistant Group of Insects

The most abundant bacterial phyla in the midgut of the selected unsusceptible EPN-exposed group of larvae were as follows: Firmicutes (48.5%), Proteobacteria (20.2%), Actinobacteria (19.4%), Bacteroidetes (6.2%), Tenericutes (1.8%), and Verrucomicrobia (1.0%) (Table 2). The midgut bacterial communities across the resistant individuals were mainly represented by the classes Clostridia (32.0%), Actinobacteria (19.4%), γ-Proteobacteria (10.1%), Bacilli (6.0%), Erysipelotrichia (8.6%), α-Proteobacteria (7.8%), and Bacteroidia (5.7%) (Table 2). The other classes detected in the midgut of the group of resistant insects exhibited RA < 1%. In the analyzed samples, 16 bacterial families were present at RA > 1%, with the greatest abundance of *Lachnospiraceae* (14.1%), *Ruminococcaceae* (13.3%), *Erysipelotrichaceae* (8.6%), *Microbacteriaceae* (6.0%), Bacteroidacae (5.1%), and *Enterobacteriacae* (4.2%) (Table 3). The most abundant genera in the midgut of the EPN-resistant insects were *Turicibacter* (8.8%), *Lachnoclostridium* (5.9%), *Bacteroidetes* (5.6%), *Anaerotruncus* (6.2%), *Ruminococcus* (3.0%), *Kineothrix* (2.4%), *Faecalicatena* (2.3%), *Enterobacter* (2.2%), and *Blautia* (2.1%) (Appendix A). The other genera displayed RA < 2%. The abundance of the detected taxa varied significantly across the samples.

### 2.6. Comparison of the Midgut Microbiota between the Control and EPN-Resistantt Group of Insects

The values of diversity indices (Shannon and Simpson), as well as bacterial community evenness indicator, did not differ significantly between the two studied groups of larvae (Appendix A). The group of the EPN-resistant insects also had a higher median of the number of bacterial genera, compared to the control larva samples (mean rank for control group 5.51 vs. mean rank for EPN-resistant group 8.41, *p* = 0.039). The PCoA and the analysis of similarity ANOSIM showed that the bacterial communities of the EPN-resistant larvae at the genus level differed significantly from those of the control larvae (ANOSIM, R = 0.15, *p* = 0.01) (Figure 1). Further analysis showed that *Lachnoclostridium, Anaerotignum, Tyzzerella, Paludicola*, and *Ruminiclostridium* were significantly more abundant in the midguts of the control group of larvae than in the EPN-resistant insects (Table 4). In contrast, *Mesorhizobium, Galbitalea, Conyzicola, Mycolicibacterium, Aeromicrobium, Herbiconiux, Cellulomonas, Friedmanniella,* and *Methylobacterium* were more abundant in the resistant group of insects. Finally, the analysis of the microbiome networks revealed that *Bacillus, Hespellia, Ruminococcus, Pseudoflavonifractor,* and *Sporobacter* were the keystone genera in the midgut of the control larvae, while *Propionispora, Herbiconiux, Conexibacter, Sporomusa*, and *Anaerotaenia* were the most interactive genera in the group of the EPN-resistant larvae (Appendix A).

### 2.7. Abundance of EPN Symbionts and Their Antagonists in the Midgut Communities

The reads assigned to the antagonistic species, such as *S. liquefaciens*, *A. calcoaceticus*, *C. murliniae*, *P. chlororaphis*, and *C. lathyri*, in total, reached RA of 0.22% and 0.19% in the control and EPN-resistant group, respectively. The most abundant antagonistic species was *S. liquefaciens* (0.14% and 0.08%), followed by *A. calcoaceticus* (0.05% and 0.07%), *C. murliniae* (0.03% and 0.04%), and *P. chlororaphis* (0.003% and <0.001%) in the control and resistant group, respectively (Figure 2, Appendix A).

*Photorhabdus* and *Xenorhabdus* bacteria were detected in all samples, reaching 0.002–0.009% and 0.003–0.021% of the total number of reads, respectively. Their sequence diversity was high, allowing separation of six *Xenorhabdus* and three *Photorhabdus* spp. with RA > 0.001%. Most sequences were attributed to *Xenorhabdus kozodoii* and *Photorhabdus luminescens* (RA 0.003% each). The RA of the *Photorhabdus* and *Xenorhabdus* genera in the control group was 0.003% and 0.008%, respectively, and 0.008% and 0.016%, respectively, in the resistant one (Appendix A).

The differences in the RA of both the EPN symbionts and the antagonistic species between the resistant and control groups were not statistically significant.

## 3. Discussion

### 3.1. Natural Gut Microbiota of the Common Cockchafer

In our study, the large amount of sequence information obtained together with the high level of accuracy of the reads revealed the complex bacterial communities associated with the midgut of *M. melolontha* larvae. The microbial diversity of the midgut of root-feeding *M. melolontha* larvae was earlier briefly studied only by Egert et al. (2005), who showed a low diversity and variable composition of the community in this gut compartment, in contrast to that of the hindgut [29]. In turn, the gut microbiota of the other closely related scarab larvae, i.e., the forest cockchafer *M. hippocastani,* has been characterized in more detail. The data showed that the larva guts exhibited a complex composition of the bacterial community, compared to that of adults, possibly reflecting adaptation to a diet shift from root feeding to that based exclusively on foliage. Interestingly, part of the gut bacterial community of *M. hippocastani* remained stable in both larvae and adult stages, and this core microbiota was composed of representatives of Proteobacteria, Bacilli, Clostridia, Erysipelotrichi, and Sphingobacteria [30].

The data described above partly coincide with the results of our studies. The Shannon diversity indices of *M. melolontha* larva midgut communities were higher than those observed for *M. hippocastani* but close to those from other *Scarabaeidae* beetles, e.g., *Pachysoma* spp. [31]. We found that approximately 93% of the bacteria found in the *M. melolontha* larva midguts represented the classes Clostridia, Actinobacteria, Bacilli, Proteobacteria, Erysipelotrichia, and Bacteroidia. A comparison of the results revealed that Clostridia and Actinobacteria were substantially more abundant in the midgut of *M. melolontha* than *M. hippocastani,* although their abundance varied significantly between individuals. In the *M. melolontha* midgut, the *Enterobacteriaceae* family was relatively abundant, in contrast to *Pseudomonadaceae,* which was scarce. Previous reports described that the wild coleopteran microbiota was usually dominated by *Enterobacteriaceae* and/or *Pseudomonadaceae*, e.g., in *M. hippocastani,* burying beetle *Nicrophorus vespiloides,* or wood-boring *Agrilus mali* [10,30,32].

The most abundant bacterial genera found in the studied communities were *Turicibacter* (Erysipelotrichia), followed by *Bacteroides Anaerotruncus*, *Lachnoclostridium, Serratia,* and *Enterococcus.* These anaerobic bacteria obviously contribute to the complex symbiont-mediated processing of root lignocellulose biomass, whose products (glucose and xylose) converted primarily into pyruvate are subsequently utilized as an energy source via the fermentation process [33]. This is corroborated by the recently described ranking of the roles of gut bacteria associated with beetle hosts, which indicates that their basic activity is essential nutrient provisioning, followed by digestion and detoxification [34].

The α-diversity analysis applied to our data suggested no significant difference between the microbial communities of the L2 and L3 larva samples. Similar results were reported in the case of *M. hippocastani*. Although some differences in the gut bacterial composition of L2 and L3 larval stages were detected using a Unifrac test, the difference in the microbiota between larval and adult stages was more evident, both in terms of α and β diversity [30]. In fact, we may not be able to detect the discrete changes in the microbial community between larvae differing in the maturity stage due to its high intra-variability. The detected high intra-individual variability of the *M. melolontha* midgut microbiota is in agreement with earlier results reported by Eger et al. (2005) [29]. While feeding on roots, scarab larvae introduce a significant amount of environmental and food bacteria, which influence their gut community composition. Nevertheless, their gut bacterial assemblies differ strongly from those in the food and soil, suggesting that host internal factors rather than external environmental factors are more crucial [4,30,35].

Elucidation of the complex interactions between core microbiota and more flexible gut residents dependent on the environment can help to understand complex soil and gut microbiomes [36,37]. In our work, besides determination of taxa richness and abundance, we investigated the *M. melolontha* midgut bacterial community using correlation networks to identify the most interactive bacterial genera. All possible keystone genera belonged to the phylum *Firmicutes*. Two of the bacterial genera exhibited relevant microbial associations, i.e., *Bacillus* and *Ruminococcus* were also more abundant, while *Hespellia, Pseudoflavonifractor*, and *Sporobacter* were in a minority, but likely participated significantly in the functioning of the studied ecosystem.

### 3.2. Alterations in Gut Microbial Composition in EPN-Resistant Insects

Our earlier experiments have established that *M. melolontha* larvae, being natural hosts for EPN, have lower susceptibility to infection than other insect, not only laboratory-reared lepidopteran *Galleria* or *Manduca*, but also wild coleopteran *Tenebrio* [24,26]. One possible explanation of this phenomenon is that larvae of these species harbor bacteria providing protection against EPN in their gut. To check the effect of midgut bacteria on host protection against EPN, we selected unsusceptible M. *melolontha* individuals after exposure of the wild larvae to a large dose of IJs. It was assumed that these individuals are resistant to EPN, preventing or reducing pathogen growth [38]. Then, the α and β diversity of the bacterial microbiota of individuals from the resistant and control groups was compared. We revealed that all of them demonstrated high α-diversity as well as intra-individual composition differences. The role of bacterial richness and diversity as a factor reflecting gut ecosystem stability and resistance to pathogens has been supported by many studies on animals, including insects [1,39,40,41].

Furthermore, we observed alteration in the microbial composition between the resistant and control groups of the insects, based on differences in the abundance of specific bacterial taxa. These changes appear to have been caused mainly by the rare community members. In the EPN-exposed unsusceptible larvae, we found a significant increase in the abundance of several bacterial genera belonging to the orders Actinomycetales and Rhizobiales, while the abundance of some other Clostridiales decreased. Additionally, the EPN-resistant insects markedly differed in the set of keystone genera from the control larvae, indicating differences in mutual interactions among the midgut-associated bacteria. There are no equivalent studies, but the investigations of *B. thuringiensis* revealed that bacterial communities of resistant insects strongly differed from susceptible ones, showing low diversity and low species richness but high inter-individual differences [42,43]. Xia et al. (2013), who examined the diversity of gut bacteria in the diamondback moth *Plutella xylostella,* found that insecticide-resistant insects hosted more Firmicutes (mainly Lactobacillales) and fewer Proteobacteria (mainly Enterobacteriales) than susceptible ones [44]. However, the gut microbiota-based effect on resistance to chemical insecticides or biological toxins of *B. thuringiensis* is expected to be different than that on resistance to EPN and their mutualistic bacteria. Additionally, such an effect may not be necessarily reflected in differences in higher level taxa, as the importance of strain-level differences and a small number of functionally active species in the complex gut microbiome were shown earlier [45,46]. Our findings suggest that a modified microbiota is associated with the higher resistance of the scarab larvae to EPN, although these results do not allow for determining whether the higher resistance of some larvae is driven, at least partially, by the gut bacteria or rather the larval immune system, while the alteration in the microbiota composition is rather a consequence of the bacterial response to EPN. Nevertheless, the PCoA analysis revealed that some insects from the control group had similar microbiota as the resistant ones, which suggests that they have altered microbiota without EPN exposure and may represent EPN-resistant individuals. This observation supports the hypothesis that microbiota already present in the gut of wild *M. melolontha* larvae play a significant role in conferring nematode resistance and observed microbiota modification is the cause of the higher resistance to EPN of some larvae rather than a consequence of nematode exposure.

On the other hand, it has been found in *B. thuringiensis* that modification of the gut microbiota of tolerant insects subjected earlier to prolonged pathogen exposure is a rapid process occurring as a consequence of infection [43]. Compositional changes in bacterial microbiota can be probably related to strong EPN-induced immune response, which trigger its disequilibrium, as reported in *B. thuringiensis* [43] or the entomopathogenic fungus *Beauveria bassiana* [47]. Changes in the microbiota composition may be also caused by high antibacterial secretory activity of EPN bacteria, which directly interact with the gut microbiota. Release of a huge amount of bacteriocins and antimicrobial secondary metabolites has been detected in *Photorhabdus* and *Xenorhabdus* bacteria [23]. Antimicrobial effector proteins have also been detected in nematodes, e.g., EPN [48], *Caenorhabditis elegans* [49] and *Ascaris suum* [50]. On the other hand, if the protective mechanisms of gut bacteria are not very specific, as in the case of bacteriocin secretion or launching the host immune system, they might have a strong ecological impact on the whole gut microbiome and a substantive shift in the microbiota structure [51,52]. Additionally, damage caused by EPN exposure may lead to changes in the gut physiology and, subsequently, alter the microbial composition [53].

### 3.3. Presence of Xenorhabdus and Photorhabdus Entomopathogens and Their Antagonists in the M. melolontha Midgut Microbiota

We previously reported that some bacterial strains present in the *M. melolontha* midgut microbiota display antagonist activity against *Xenorhabdus* and *Photorhabdus* spp., limiting their growth in vitro, which supports the concept of competition between EPN bacteria and gut microbiota [24]. These antagonistic strains represent the following species: *S. liquefaciens*, *A. calcoaceticus*, *C. murliniae*, *P. chlororaphis*, and *C. lathyri.* These bacteria are rare community members. However, the abundance of *S. liquefaciens* and *A. calcoaceticus* reached a relatively high level in some of the samples, i.e., up to 0.3 and 0.1%, respectively, while *C. lathyri* was characterized by the lowest frequency, i.e., <0.001%. Nevertheless, we did not observe a higher frequency of these bacterial species in the midgut of the unsusceptible EPN-exposed larvae in comparison with the control ones. Hence, the role of the isolated earlier gut-associated antagonists in outcompeting EPN bacteria in nature should still be investigated. These strains probably display broad bioactive compound secretory activity, targeted not only to *Xenorhabdus* and *Photorhabdus* bacteria. Additionally, the secretion of bioactive molecules against entomopathogens may be strain-specific, discrete, and dependent on corresponding metabolites [45,54,55].

Additionally, we detected a low number of reads assigned to *Xenorhabdus* and *Photorhabdus* in all midgut samples. The analysis of the *Xenorhabdus* and *Photorhabdus* spp. showed no significant increase in their abundance in the midgut of the resistant individuals, compared to the control ones. This suggests that *Xenorhabdus* and *Photorhabdus* bacteria were already present in the midguts of the larvae when they were collected from the environment, as they are rare but widespread gut residents of insect larvae living in soil. To date, there is little information on this phenomenon; however, *Photorhabdus* spp. was detected earlier in laboratory-reared *T. molitor* [56]. This is an interesting observation, as EPN bacteria have never been isolated from soil and are generally regarded to be obligate symbionts of nematodes [22]. A unique feature of *Xenorhabdus* and *Photorhabdus* bacteria is the existence in two different phenotypic forms, i.e., 1 and 2 cells, from which only the primary form is able to live in symbiosis with nematodes. The biological function of secondary form cells is unclear, but recent findings indicate that they remain in soil and live freely [57]. Since they are easily ingested by insect larvae, it is conceivable that they can also easily adapt to such an alternative environment as the insect gut.

## 4. Materials and Methods

### 4.1. Sample Collection and Preparation

*M. melolontha* L2 and L3 larvae were sampled over a short period in different areas of the Lublin region (Eastern Poland) in April 2019 (Table 1). The grubs collected randomly from soil were placed separately in vented plastic cups with soil taken from the same field and immediately transported to the laboratory for analysis of the species and stage of development. In total, over 200 healthy *M. melolontha* second- and third-instar larvae were selected for further study. They were maintained for 3 days in 20-mL sterilized soil samples to exclude individuals that were invisibly wounded or infected with pathogens. Next, 30 randomly selected larvae referred to as “wild” or “control” individuals were subjected to 1-day starving (to limit the risk of contamination of extracted gut microbiota DNA by that coming from the larval alimentary bolus) and then surface sterilized with 70% alcohol, washed twice in sterile distilled water, and allowed to dry for 1 min. Subsequently, each of these larvae were decapitated and the digestive tract was dissected (Phot. S1). After separation, the midgut was placed in an Eppendorf tube and frozen at −85 °C. Simultaneously, the remaining larvae were exposed to EPN in a dose of 1000 IJ per 1 insect according the standard procedure [58]. We used 5 different EPN species from our laboratory collection: *Hetorhabditis megidis, Steinernema arenarium, Steinernema bicornutum, Steinernema carpocapsae,* and *Steinernema kraussei* to infest 30 larvae with each EPN species tested. The cups with the insects were kept in incubators at 20 °C for 12 days, allowing infection. Afterwards, moving and healthy looking larvae were qualified as “EPN-resistant” and subjected to gut dissection as described above after 1-day starving. Additionally, the survival ratio was calculated as the proportion of alive larvae obtained after the exposure to each EPN species used. Insects were considered dead if they did not move even after being touched with the preparation needle. A schematic diagram of the course of the experiment is shown in Appendix A.

### 4.2. DNA Extraction

Total microbial community DNA was extracted from the dissected *M. melolontha* midgut using a Bead-Beat Micro AX Gravity kit (A&A Biotechnology, Gdynia, Poland). The DNA concentration was measured with Qubit 2.0 (Invitrogen, Carlsbad, CA, USA) using a Qubit dsDNA HS Assay Kit (Thermo Fisher Scientific, Waltham, MA, USA). Extraction of environmental DNA, next-generation sequencing, and basic bioinformatic analysis were performed by genXone (Złotniki, Poland).

### 4.3. Nanopore High Throughput Sequencing

The hyper-variable regions of the 16S rRNA bacterial gene (V3-V8) were amplified with primers F338 5′-ACT CCT ACG GGA GGC AGC-3′ [59] and R1391 5′-GACGGGCGGTGTGTRCA-3′ [60]. Bacterial libraries were created using a Ligation Sequencing Kit 1D and sequenced on a GridION X5 sequencer (Oxford Nanopore Technologies, Oxford, UK). The reads obtained were filtered for their quality (average read quality ≥ 10 Phred quality score, the average quality of read fragments with a window size of 500 bp ≥ 6 Phred score) and length (minimum 800 bp). During the demultiplexing step, the removal of adapters and barcodes was performed by Porechop v.0.2.4. High-quality reads were processed for taxonomic identification by matching the NGS sequences with sequences deposited in the NCBI using a modified BLAST algorithm.

### 4.4. Exploratory Data Analyses

The subsampling for the obtained reads at the genus level was performed using phyloseq in R (ver. 4.1.0) for 1000-fold randomized subsamples of 72,721 sequences per sample. The rarefraction curves for the obtained reads as well as the diversity indices, such as Shannon–Wienner, Simpson (1-D), evenness indicator (Shannon evenness), and Principal Component Analysis (PCoA), were calculated and created using a vegan package [61] in R (ver. 4.1.0). The importance of the differences in the diversity indicators between the EPN-resistant larvae and the control group was assessed using the non-parametric Mann–Whitney U test in Past v 3.08 [62]. The analysis of similarities ANOSIM was applied to test differences at the genus level in the community composition between the group of EPN-resistant individuals and the control one. The importance of intergroup differences in the RA of dominant genera, as well as EPN bacteria/antagonistic species, was checked with the Mann–Whitney U test. Microbiome networks constrained to the 100 most abundant genera in both groups of samples were built using the igraph, qgraph, vegan, and MCL packages in R, as recommended by Layeghifard et al. (2018) [63]. The microbiome networks were made using the Sparse Correlations for Compositional data (SparCC) method proposed by Friedman and Alm (2012) [64]. Keystone taxa were detected using the link-analysis method [63]. Heatmaps presenting the bacteria *Xenorhabdus* and *Photorhabdus* spp. and their antagonists, such as *Acinetobacter calcoacticus, Chryseobacterium lathyri, Citrobacter murliniae, Pseudomonas chlororaphis*, and *Serratia liquefaciens* were prepared using a pheatmap in R. Prior, raw data for the analysis were log-base 10 transformed.

Sequencing data were deposited in NCBI under accession number PRJNA665354.

## 5. Conclusions

Our study based on metataxonomic nanopore-sequencing analysis provides detailed information about the diversity of the bacterial midgut community of *M. melolontha* beetle larvae. To study of the role of the gut microbiota in the protection of insect hosts against EPN infection, we exposed *M. melolontha* larvae to a large dose of this pathogen and compared the microbiota of resistant individuals with that of the control insects. The midgut community of EPN-resistant insect larvae showed significant differences in the abundance of many bacterial genera, including sets of keystone bacterial genera. However, whether the modified microbiota is a cause or a consequence of the exposure of larvae to EPN remains an open question. An alternative approach to test the contribution of microbiota in resistance to pathogens is the elimination of bacteria from the insect gut using antibiotics; however, a disadvantage of this approach is the direct effect of antibiotics on the insect physiology. In general, to speculate how the changes in the gut microbiome drive the course of EPN pathogenesis, wider studies are needed, including gene functionality analysis as well as controlled microbiome manipulation. Elucidation of the tripartite interactions between insect immunity, EPN infection, and host gut microbiota is necessary for identification of factors determining the outcome of infections. This may offer great potential for improvement of methods for control of harmful insects.

## Figures and Tables

**Figure 1 pathogens-10-00396-f001:**
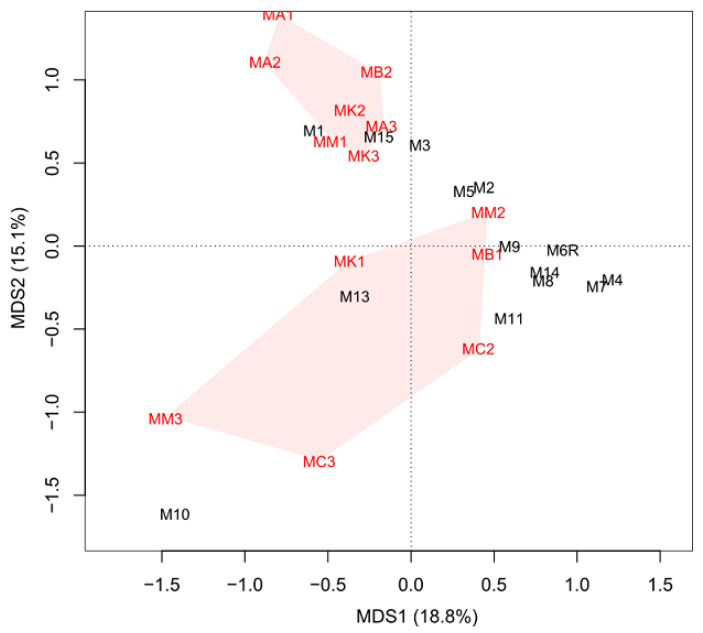
Principal component analysis (PCoA) of the reads obtained from nanopore sequencing of the 16S rRNA gene. M—individuals of *M. melolontha* from the control group; MA—individuals exposed to *S. arenarium*; MB—*S. bicornutum*; MC—*S. carposapsae*; MK—*S. kraussei*; MM—*H. megidis*; the figure refers to the serial number of the sample.

**Figure 2 pathogens-10-00396-f002:**
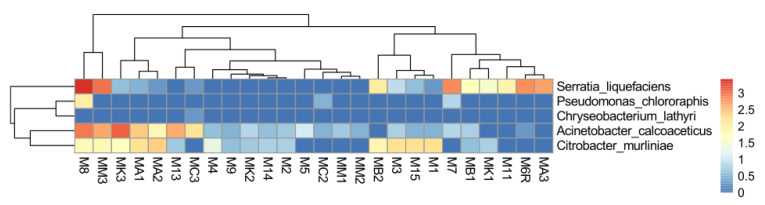
Heatmap showing the relative abundance of bacterial species exhibiting antagonistic activity against EPN bacteria in the midguts of all tested individuals. M—individuals of *M. melolontha* from the control group; MA—individuals exposed to *S. arenarium*; MB—*S. bicornutum*; MC—*S. carposapsae*; MK—*S. kraussei*; MM—*H. megidis*; the figure refers to the serial number of the sample.

**Table 1 pathogens-10-00396-t001:** List of sampling sites.

Sample Name	Developmental Stage of Larvae	Geographic Origin	Geographic Origin Code	EPN Exposure/EPN (Bacterial Symbiont) Species
M1, M2	L2	Forest 50°54′41.9″ N 22°19′27.1″ E	FK	none
M3, M4	L3	Forest 50°54′41.9″ N 22°19′27.1″ E	FK	none
M5, M6	L3	Forest nursery 51°23′37.0″ N 22°29′44.0″ E	KF	none
M7	L3	Forest nursery 51°23′37.0″ N 22°29′44.0″ E	KF	none
M8, M9	L2	Forest nursery 51°23′37.0″ N 22°29′44.0″ E	KF	none
M10, M11, M12	L2	Forest 51°23′45.9″ N 22°47′43.2″ E	ZF	none
M13, M14	L2	Field 51°18′44.8″ N 22°24′33.7″ E	PF	none
M15	L3	Field 51°18′44.8″ N 22°24′33.7″ E	PF	none
MM1, MM2	L2	Forest nursery 51°23′37.0″ N 22°29′44.0″ E	KF	*Heterorhabditis megidis* (*Photorhabdus temperata*)
MM3	L3	Forest nursery 51°23′37.0″ N 22°29′44.0″ E	KF	*Heterorhabditis megidis* (*Photorhabdus temperata*)
MA1, MA2, MA3	L2	Forest nursery 51°23′37.0″ N 22°29′44.0″ E	KF	*Steinernema arenarium* (*Xenorhabdus kozodoii)*
MC1, MC2	L2	Forest nursery 51°23′37.0″ N 22°29′44.0″ E	KF	*Steinernema carpocapse* (*Xenorhabdus nematophila*)
MC3	L3	Forest nursery 51°23′37.0″ N 22°29′44.0″ E	KF	*Steinernema carpocapsae* (*Xenorhabdus nematophila*)
MB1, MB2	L2	Forest nursery 51°23′37.0″ N 22°29′44.0″ E	KF	*Steinernema bicornutum* (*Xenorhabdus budapestensis*)
MB3	L3	Forest nursery 51°23′37.0″ N 22°29′44.0″ E	KF	*Steinernema bicornutum* (*Xenorhabdus budapestensis*)
MK1, MK2	L2	Forest nursery 51°23′37.0″ N 22°29′44.0″ E	KF	*Steinernema kraussei* (*Xenorhabdus bovienii)*
MK3	L3	Forest nursery 51°23′37.0″ N 22°29′44.0″ E	KF	*Steinernema kraussei* (*Xenorhabdus bovienii)*

**Table 2 pathogens-10-00396-t002:** Composition of midgut-associated bacteria in the *M. melolontha* larvae at the phylum and class level.

Taxname	Relative Abundance (Mean in %) and 95% Confidence Interval	
	Control Larvae	EPN-Resistant Larvae
**Phylum level**		
Firmicutes	72.85 [63.16–82.54]	48.45 [33.59–63.31]
Proteobacteria	12.15 [5.61–18.69]	20.17 [12.68–27.65]
Actinobacteria	5.85 [2.42–9.27]	19.35 [10.74–27.96]
Bacteroidetes	6.75 [3.03–10.47]	6.18 [2.33–10.03]
Tenericutes	0.32 [0.05–0.58]	1.77 [0.10–3.43]
Verrucomicrobia	0.26 [0.02–0.50]	1.01 [0.34–61.68]
**Class level**		
Clostridia	49.43 [33.29–65.58]	31.95 [18.85–45.06]
Actinobacteria	5.85 [2.42–9.27]	19.35 [10.74–27.96]
Bacilli	12.74 [4.21–21.27]	5.99 [3.44–8.53]
γ-Proteobacteria	6.89 [1.70–12.08]	10.09 [3.67–16.50]
Erysipelotrichia	7.81 [0–19.04]	8.54 [0–17.65]
Bacteroidia	6.55 [2.74–10.36]	5.70 [1.82–9.57]
α-Proteobacteria	3.14 [0.03–6.24]	7.75 [3.95–11.54]
β-Proteobacteria	1.07 [0.32–1.81]	1.01 [0–2.07]
Mollicutes	0.32 [0.05–0.58]	1.77 [0.10–3.43]

Main abundant phyla and classes (>1%) are indicated.

**Table 3 pathogens-10-00396-t003:** Composition of midgut-associated bacteria in the *M. melolontha* larvae at the family level.

Taxname	Relative Abundance (Mean in %) and 95% Confidence Interval	
	Control Larvae	EPN-Resistant Larvae
*Lachnospiraceae*	21.44 [11.31–31.56]	14.11 [5.80–22.42]
*Ruminococcaceae*	18.04 [10.72–25.36]	13.27 [2.47–24.07]
*Erysipelotrichaceae*	7.81 [0.25–19.04]	8.54 [0–17.56]
*Bacteroidacae*	4.95 [1.79–8.10]	5.08 [1.28–8.89]
*Microbacteriaceae*	1.37 [0.19–2.55]	6.00 [2.23–9.76]
*Enterobacteriaceae*	2.12 [0–4.85]	4.19 [0.23–8.16]
*Bacillaceae*	4.13 [0.95–7.30]	1.88 [1.08–2.69]
*Sporomusaceae*	2.00 [0.64–3.07]	1.42 [0.56–2.28]
*Nocardioidaceae*	0.39 [0.13–0.66]	1.35 [0–2.86]
*Hungateiclostridiaceae*	2.54 [1.12–3.95]	0.36 [0.24–0.49]
*Clostridiaceae*	1.70 [1.04–2.36]	1.02 [0.58–1.46]
*Staphylococcaceae*	1.47 [0–3.64]	1.02 [0.24–1.80]
*Rhizobiaceae*	1.01 [0–2.62]	1.35 [0.62–2.09]
*Enterococcaceae*	1.52 [0–3.77]	0.70 [0–1.43]
*Propionibacteriaceae*	0.58 [0.02–1.13]	1.52 [0–3.22]
*Morganellaceae*	1.91 [0–5.98]	0.05 [0.01–0.10]
*Erwiniaceae*	0.28 [0.01–0.54]	1.75 [0–3.72]
*Sphingomonadaceae*	0.44 [0–1.14]	1.56 [0.6–2.52]
*Bradyrhizobiaceae*	0.68 [0.19–1.17]	1.22 [0.53–1.91]

Main abundant families (>1%) are indicated.

**Table 4 pathogens-10-00396-t004:** Statistical differences between the most abundant bacterial genera in the control insects and the entomopathogenic nematodes (EPN)-resistant group of insects.

	Relative Abundance Median		Mean Rank		Mann-Whitney U	*p*-Value
	Control *n* = 14	EPN-Resistant *n* = 13	Control *n* = 14	EPN-Resistant *n* = 13		
*Lachnoclostridium*	2.028	0.956	8.92	5.07	46	0.029
*Anaerotignum*	1.850	0.264	9.59	4.01	28	0.001
*Tyzzerella*	1.256	0.085	9.48	4.51	31	0.004
*Paludicola*	0.492	0.169	8.81	5.19	49	0.044
*Ruminiclostridium*	0.325	0.051	9.19	4.81	39	0.010
*Mesorhizobium*	0.019	0.061	5.46	8.54	42.5	0.019
*Galbitalea*	0.001	0.139	5.22	8.78	36	0.007
*Conyzicola*	0.002	0.056	4.96	9.04	29	0.002
*Mycolicibacterium*	0.054	0.360	5.37	8.63	40	0.013
*Aeromicrobium*	0.007	0.232	5.29	8.70	38	0.009
*Herbiconiux*	0.003	0.162	5.04	8.96	31	0.003
*Cellulomonas*	0.003	0.223	5.20	8.80	35.5	0.005
*Friedmanniella*	0.001	0.164	4.59	9.41	19	0.0003
*Methylobacterium*	0.006	0.076	5.61	8.39	46.5	0.030

## Data Availability

Not applicable.

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
