# Peer review of "Nanopore-Sequencing Characterization of the Gut Microbiota of Melolontha melolontha Larvae: Contribution to Protection against Entomopathogenic Nematodes?"

_pathogens, 2021, doi:10.3390/pathogens10040396_

Round 1

Reviewer 1 Report

The work presented in the ms entitled  “Nanopore-sequencing characterization of the gut microbiota of Melolontha melolontha larvae: contribution to protection against entomopathogenic nematodes?” is very novel and very interesting. Highlighting the fact that insect resistance to entomopathogenic nematodes may be linked to the structure of their intestinal bacterial community, is very essential for future studies. I found the work to be very well-organized and presented; the authors used all the proper statistical analyses to analyze and interpret their data set. The introduction is making a strong case, highlighting the need for this study, and the objectives of the study are clearly given. The authors further provided as supplementary material all data need to support their findings.

  1. The abstract must be re-written. The most significant results of this study must be included. The way it is written now does not include most of the important information that can be found in the text.
  2. I found the discussion to be very long. The length must be reduced, as many parts of the discussion can be omitted. Too much detail has been added, and this is not helpful for the reader as the major points cannot be clearly highlighted  
  3. The conclusion section must also be re-written. In this section, the authors must not try and present once again the need for the study. They must present the most important findings of their study. This part along with the abstract were the weak parts of this ms
  4. The legends of Figures 1 and 2 should contain all the necessary information for the reader to understand the graph. In order to follow them both, I had to check other parts of the ms, and this is very annoying for the reader

Author Response

Dear Reviewer,

We would like to thank you for the useful comments to improve the paper. We have addressed all the comments as explained below.

  1. The abstract must be re-written

We have re-written the abstract according to the reviewer’s suggestion. Now it better summarizes the paper and includes the most important results of the study.

  1. I found the discussion very long.

The length of the discussion section has been reduced. In each subsection of the discussion, some inessential information has been deleted. In total, 35 lines of the text in the Discussion have been removed, along with 12 references.

  1. The conclusion section must be re-written

We have re-written the conclusion section according to the reviewer’s suggestion. Now it includes the most important findings of the study and future perspective.

  1. The legends of Figures 1 and 2 should contain all the necessary information to understand the graph.

Explanations of the letter and figure abbreviations in figures 1 and 2 have been added.

“M – individuals of M. melolontha from the control group; MA – individuals exposed to S. arenarium; MB - S. bicornutum; MC - S. carposapsae; MK - S. kraussei; MM - H. megidis; the figure refers to the serial number of the sample”

Reviewer 2 Report

This study by Sajnaga et al is focused on the potential relationship between gut microbiota in Melolontha melolontha larvae and their tolerance to EPN infection. They used Nanopore sequencing to identify the diversity and abundance of bacterial species in the midgut of M. melolontha, utilizing control larvae and unsusceptible larvae. Overall, the study is interesting and the data will be helpful to the field.

Major Issues:

One major issue with this paper is the fact that hosts that did not die when exposed to 1000 EPN IJs are subsequently referred to as “EPN-tolerant” hosts. The authors have missed the important distinction between resistance and tolerance, where resistance represents the ability of the host to prevent or reduce pathogen growth, and tolerance, which is the ability of the host to tolerate the presence of the pathogen while minimizing pathology (Ayres and Schneider 2012; Louie et al. 2016; Råberg et al. 2009; Råberg et al. 2007; Schneider and Ayres 2008). The authors have not determined whether the insects are resistant or tolerant to the presence of the EPNs. However, it seems much more likely that the EPNs are either killed by the host immune response or not able to infect these hosts very well. Thus, it seems more likely that the insects are resistant to the EPNs, not tolerant of them. This language should be corrected in the manuscript.

The second major issue with the paper is the fact that the methodology precludes any conclusions about cause and effect. They convincingly demonstrate that the bacteria in the insect gut are different between the control and the EPN-resistant hosts. But they cannot determine whether the difference was present in those insects before they were infected or whether that change was driven by the exposure to the EPNs. The authors seem to suggest that the microbiota of the EPN-resistant insects had evolved to protect them from infection. This language should be softened, since no solid cause and effect conclusion can be drawn. The authors apparently exposed 170 larval insects to the EPNs, and that most of the insects were resistant to even 1,000 IJs, with S. arenarium appearing to be the most deadly, killing 55% of exposed hosts. But compared to other insects, M. melolontha are quite resistant to EPNs, and the microbiota that are already present may play a significant role in this. For example, M1, M15, M13 all appear to have similar PCoA analyses results as EPN-resistant insects for components 1 and 2 (Figure 1). Doesn’t this suggest that some control insect hosts that were not previously exposed to the EPNs may already have microbiota similar to the insects that are EPN-resistant?

A straightforward experiment to test the contribution of the microbiota may simply be feeding antibiotics to some of the larvae prior to EPN exposure. The authors would also need to evaluate the efficacy of the antibiotics using proper controls.

Minor Issues:

There are some typos and language issues. Some of those are highlighted below:

Abstract: correct “unsuspectible” to “unsusceptible.”

Page 2 second to last paragraph. Change to “to address the question of whether the differences…” Also, remove the extra spaces between “isolated” and “earlier.”

Page 3, change to “of the larvae were investigated using fourth-generation…”

Page 4, correct the spelling of “relative abundance,” in section 2.3.

Page 5, change to “the effect of the larval developmental stage…”

Page 9, bottom of the page, change to “Species-rich gut communities or those with a high…”

Page 11, 3.3, change to “These bacteria represented rare community members…”

Page 13, Conclusion, change to “This study sheds light…”

References:

Ayres JS, Schneider DS (2012) Tolerance of infections Annual review of immunology 30:271-294 doi:10.1146/annurev-immunol-020711-075030

Louie A, Song KH, Hotson A, Tate AT, Schneider DS (2016) How many parameters does it take to describe disease tolerance? Plos Biology 14 doi:10.1371/journal.pbio.1002435

Råberg L, Graham AL, Read AF (2009) Decomposing health: tolerance and resistance to parasites in animals Philosophical transactions of the Royal Society of London Series B, Biological sciences 364:37-49 doi:10.1098/rstb.2008.0184

Råberg L, Sim D, Read AF (2007) Disentangling genetic variation for resistance and tolerance to infectious diseases in animals Science 318:812-814 doi:10.1126/science.1148526

Schneider DS, Ayres JS (2008) Two ways to survive infection: what resistance and tolerance can teach us about treating infectious diseases Nature reviews Immunology 8:889-895 doi:10.1038/nri2432

Author Response

Dear Reviewer,

We would like to thank you for the useful comments to improve the paper. We have addressed all the comments as explained below.

  1. The authors have missed the important distinction between resistance and tolerance…This language should be corrected in the manuscript.

This remark is very valuable to us because we are just starting research in the field of potential relationships between gut microbiota of insect larvae and their resistance to entomopathogenic nematode infection. We have changed “EPN-tolerant” to “EPN-resistant” and “EPN tolerance” to “EPN resistance” in the manuscript and supplementary data. Additionally, we have included a short note on this issue in the Discussion section along with an appropriate reference.

“To check the effect of midgut bacteria on host protection against EPN, we selected unsusceptible Mmelolontha individuals after exposure of the wild larvae to a large dose of IJ. It was assumed that these individuals are resistant to EPN, preventing or reducing pathogen growth [38]”.

Råberg, L., Sim, D., Read, A.F. Disentangling genetic variation for resistance and tolerance to infectious diseases in animals. Science 2007, 318, 812-814. doi: 10.1126/science.114852

2. The authors seem to suggest that the microbiota of the EPN-resistant insects had evolved to protect them from infection…This language should be softened since no solid cause and effect conclusion can be drawn.

We agree that the methodology used in our study precludes any conclusion about the cause and effect. To avoid any misunderstanding of this issue, the sentence: “Insects facing EPN/mutualistic bacteria complexes for ca. 200-500 millions of years co-evolved with them, allowing the development of different resistance mechanisms, probably including a selection of gut microbes toward optimized protection of the insect host” has been removed. In addition, to make it clear, there is a statement in the paper: “Our findings suggest that a modified microbiota is associated with the higher resistance of the scarab larvae to EPN, although these results do not allow determining whether the higher resistance of some larvae is driven, at least partially, by the gut bacteria or rather the larval immune system, while the alteration in the microbiota composition is rather a consequence of the bacterial response to EPN” along with a detailed discussion on this topic. Finally, in the conclusions, we have added:” However, whether the modified microbiota is a cause or a consequence of the exposure of larvae to EPN remains an open question”.

  1. A straightforward experiment to test the contribution of microbiota may simply be feeding antibiotics to some of the larvae prior to EPN exposure. The authors would also need to evaluate the efficiency of the antibiotics using proper controls.

Indeed, the role of microbiota in the output of pathogen infection has been investigated in many insect species, mostly lepidopterans, using antibiotics to eliminate the bacteria from the insect. Nevertheless, this approach is a bit controversial, as it strongly disturbs the physiology of insects.  In our study, we provide an alternative approach to the study of the contribution of microbiota in pathogen infection by exposing the larvae to a large dose of entomopathogenic nematodes and comparing the microbiota of the most resistant individuals with that of control ones. However, to highlight that the antibiotic approach may provide new information on the subject, we have added a fragment about it in the conclusion section.

“An alternative approach to test the contribution of microbiota in resistance to pathogens is the elimination of bacteria from the insect gut using antibiotics; however, a disadvantage of this approach is the direct effect of antibiotics on the insect physiology.”

  1. Reviewer 2 mentioned several typos and grammatical errors.

We have fixed all of them. We have also proofread the paper to eliminate such errors.

Abstract: correct “unsuspectible” to “unsusceptible.” - corrected

Page 2 second to the last paragraph. Change to “to address the question of whether the differences…” Also, remove the extra spaces between “isolated” and “earlier.” - corrected

Page 3, change to “of the larvae were investigated using fourth-generation…” - corrected

Page 4, correct the spelling of “relative abundance,” in section 2.3. - corrected

Page 5, change to “the effect of the larval developmental stage…” - corrected

Page 9, bottom of the page, change to “Species-rich gut communities or those with a high…”- this sentence has been removed from the manuscript, as the discussion has been shortened according to the recommendation from Reviewer 1

Page 11, 3.3, change to “These bacteria represented rare community members…” - to avoid repeating the word “represent”, we decided to change “were constituted” to “are”

“These antagonistic strains represent the following species: S. liquefaciens, A. calcoaceticus, C. murliniae, P. chlororaphis, and C. lathyri. These bacteria are rare community members”

Page 13, Conclusion, change to “This study sheds light…” - this sentence has been removed from the manuscript, as the conclusion has been re-written according to the recommendation from Reviewer 1.